# Exhaled Nitric Oxide Reflects the Immune Reactions of the Airways in Early Rheumatoid Arthritis

**DOI:** 10.3390/biomedicines12050964

**Published:** 2024-04-26

**Authors:** Tomas Weitoft, Johan Rönnelid, Anders Lind, Charlotte de Vries, Anders Larsson, Barbara Potempa, Jan Potempa, Alf Kastbom, Klara Martinsson, Karin Lundberg, Marieann Högman

**Affiliations:** 1Centre for Research and Development, Uppsala University, Region Gävleborg, 801 88 Gävle, Sweden; anders.lind@regiongavleborg.se; 2Rheumatology, Department of Medical Science, Uppsala University, 751 85 Uppsala, Sweden; 3Department of Immunology, Genetics and Pathology, Uppsala University, 751 85 Uppsala, Sweden; johan.ronnelid@igp.uu.se; 4Rheumatology Unit, Department of Medicine, Karolinska University Hospital, 171 76 Solna, Sweden; charlotte.de.vries@ki.se (C.d.V.); karin.lundberg@ki.se (K.L.); 5Clinical Chemistry, Department of Medical Science, Uppsala University, 751 85 Uppsala, Sweden; anders.larsson@akademiska.se; 6Department of Oral Immunity and Infectious Diseases, School of Dentistry, University of Louisville, 501 S. Preston St., Louisville, KY 40202, USA; barbara.potempa@louisville.edu (B.P.); jan.potempa@icloud.com (J.P.); 7Faculty of Biochemistry, Biophysics and Biotechnology, Jagiellonian University, Gronostajowa St. 7, 31-387 Krakow, Poland; 8Department of Biomedical and Clinical Sciences, Linköping University, 581 83 Linköping, Sweden; alf.kastbom@liu.se (A.K.); klara.martinsson@liu.se (K.M.); 9Department of Medical Science, Respiratory, Allergy and Sleep Research, Uppsala University, 751 85 Uppsala, Sweden; marieann.hogman@uu.se

**Keywords:** rheumatoid arthritis, free secretory component, ACPA, exhaled nitric oxide, lung, pathogenesis, rheumatoid factor

## Abstract

Patients with rheumatoid arthritis (RA) have altered levels of exhaled nitric oxide (NO) compared with healthy controls. Here, we investigated whether the clinical features of and immunological factors in RA pathogenesis could be linked to the NO lung dynamics in early disease. A total of 44 patients with early RA and anti-citrullinated peptide antibodies (ACPAs), specified as cyclic citrullinated peptide 2 (CCP2), were included. Their exhaled NO levels were measured, and the alveolar concentration, the airway compartment diffusing capacity and the airway wall concentration of NO were estimated using the Högman–Meriläinen algorithm. The disease activity was measured using the Disease Activity Score for 28 joints. Serum samples were analysed for anti-CCP2, rheumatoid factor, free secretory component, secretory component containing ACPAs, antibodies against *Porphyromonas gingivalis* (Rgp) and total levels of IgA, IgA1 and IgA2. Significant negative correlations were found between the airway wall concentration of NO and the number of swollen joints (Rho −0.48, *p* = 0.004), between the airway wall concentration of NO and IgA rheumatoid factor (Rho −0.41, *p* = 0.017), between the alveolar concentration and free secretory component (Rho −0.35, *p* = 0.023) and between the alveolar concentration and C-reactive protein (Rho −0.36, *p* = 0.016), but none were found for anti-CCP2, IgM rheumatoid factor or the anti-Rgp levels. In conclusion, altered NO levels, particularly its production in the airway walls, may have a role in the pathogenesis of ACPA-positive RA.

## 1. Introduction

Rheumatoid arthritis (RA) is a systemic inflammatory disease mainly affecting the joints and may lead to joint destruction and disability. In 50–80% of cases, autoantibodies against citrullinated proteins (ACPAs) or the Fc part of immunoglobulin G (IgG), rheumatoid factor (RF), are found [1]. ACPAs probably have an essential role in the disease, as they are also associated with a more severe disease course [2,3]. Consequently, RA with ACPAs and without ACPAs are often regarded as two different disease entities and may have different aetiologies [4].

Theories on the aetiology of RA involve a genetic predisposition and triggering exogenous factors, such as smoking [5], which may lead to a posttranslational shift in the amino acid arginine to citrulline in specific proteins, induced by the enzyme peptidyl arginine deaminase (PAD) [6]. This immune reaction may be triggered by inhaled agents and start in the mucosa of the gums, airways or lungs. The locally produced secretory antibodies (IgA and IgM) are transported through the mucosa by the polymeric immunoglobulin (poly Ig) receptor [7]. A shredded part of this receptor, free secretory component (free SC), is elevated in the serum of patients with ACPA-positive RA already before arthritis onset [8].

Another theory includes the bacterium *Porphyromonas gingivalis*, involved in periodontitis disease, which also expresses a PAD enzyme which may induce protein citrullination [9]. In accordance, RA patients have increased anti-*P. gingivalis* antibody levels, even before arthritis onset [10], and RA is four times more frequent in patients with periodontitis compared to the general population [11].

Nitric oxide (NO) is an important molecule in the inflammation process, inducing vascular dilatation and permeability [12], and is relevant in the cellular reactions of oxidative stress [13]. Elevated levels are found in the serum and synovial fluid of patients with RA [14]. In exhaled gas, the measured NO is produced by the cells of the airways and alveoli but is also influenced by the capillary diffusion of NO in the airways. In an extended NO analysis, using multiple NO measurements at different exhaled flows, it is possible to determine in which part of the lungs NO is produced. The Högman–Meriläinen algorithm (HMA) [15] gives estimates of the alveolar concentration (CA_NO_), the airway compartment diffusing capacity (Daw_NO_) and the airway wall concentration of NO (Caw_NO_). Altered levels are found in disorders with inflammatory changes in the lungs and airways, such as asthma and chronic obstructive pulmonary disease [14], but also in rheumatic diseases, such as Sjögren’s disease [16] and systemic sclerosis [17].

In a previous cross-sectional study on patients with chronic RA, we found not only lower NO levels in terms of CA_NO_ and Caw_NO_ but also a higher Daw_NO_ when compared with matched healthy controls [18]. Similar results were found in ACPA-positive RA patients with early disease investigated before any treatments were initiated, suggesting that these changes may reflect pulmonary involvement in its pathogenesis [19]. These subjects, representing a homogenous autoimmune disease entity, all with ACPAs, were re-analysed in the present study, where the objectives were to elucidate whether their exhaled NO levels were associated with characteristic RA autoantibodies and other markers of the autoimmune process.

## 2. Materials and Methods

### 2.1. Study Population

Patients (n = 51) with recent-onset ACPA-positive RA according to the 2010 classification criteria [20] were recruited at diagnosis on their first visit to the rheumatology department at Gävle Hospital, Sweden. After their informed consent and physical examination by a rheumatologist, the patients were included in the study. Patients with a symptom duration of more than two years at diagnosis, those treated with >10 mg prednisolone and those with difficulties understanding the study information were excluded (n = 7). The disease activity was measured using the Disease Activity Score for 28 joints (DAS28) [21], and disability was assessed according to the Health Assessment Questionnaire (HAQ) [22].

### 2.2. NO Analysis

The NO measurements were performed in accordance with the 2005 American Thoracic Society [23] and the European Respiratory Society [24]. The exhaled NO was analysed at an exhalation flow of 50 mL/s (FE_NO,50_) using an EcoMedics DLC 88 (Eco Medics AG, Dürnten, Switzerland). The NO parameters were calculated using the nonlinear HMA method with exhalation target flows of 20, 100 and 300 mL/s or using the Tsoukias and George method with flows of 100, 200 and 300 mL/s [24] when the participants could not perform the lowest flow of 20 mL/s. A constant exhaled flow was facilitated using flow resistors, and a visual feedback system guided the patients in maintaining the targeted flow throughout the exhalation. The exact flow was measured. A calculated FE_NO,50_ value for the HMA was derived for each subject and compared with the measured value as quality control [24]. The HMA method estimates the NO parameters CA_NO_, Caw_NO_ and Daw_NO_.

Serum nitrate/nitrite (NOx) was analysed using a Cayman nitrate/nitrite colorimetric assay kit (Ann Arbor, MI, USA). The total coefficient of variation for the NOx assay was 3.4%, and the detection limit was 1 µM/L.

### 2.3. Spirometry

Pre-bronchodilator spirometry was performed after the NO analysis using a Welch Allyn Spiro Perfect II (Welch Allyn, Skaneateles Falls, NY, USA). The reference values are presented as the percentages predicted using the Swedish reference values [25,26].

### 2.4. Blood Analyses

Samples were collected for analysis of their inflammatory markers, such as the erythrocyte sedimentation rate (ESR), C-reactive protein (CRP) and autoantibodies, including RF (IgA and IgM) and ACPAs (anti-CCP2 IgA and IgG), as well as free secretory component (SC), secretory component containing ACPAs (SC ACPAs) and antibodies against the *P. gingivalis* virulence factor arginine gingipain (Rgp) as a marker for periodontal infection/periodontitis.

### 2.5. Analyses of ACPA and RF

IgG and IgA anti-CCP2 and IgA and IgM RF were analysed using a fluorescence enzyme immunoassay (Elia, Thermo Fischer Scientific, Uppsala, Sweden) and using a Phadia 250 instrument (Thermo Fisher Scientific) according to the manufacturer’s instructions. The cut-off levels for anti-CCP2 IgG and IgA were 7 arbitrary units (AU). For RF IgM, the cut-off was 5 AU, and for RF IgA, it was 20 AU, as suggested by the manufacturer.

### 2.6. Analyses of Free SC, SC ACPAs, Anti-Rgp IgG and Total IgA, IgA1 and IgA2

Free SC was analysed using an in-house sandwich ELISA [8,27]. Briefly, the serum samples were diluted 1:25, added to microtiter plates pre-coated overnight with 10 μg/mL of anti-free SC 6B3 monoclonal antibody (mAb) and incubated at 37 °C for 90 min for the serum and 60 min for the detection antibody. Following washing, an HRP-conjugated anti-SC mAb 5D8, diluted 1:100, was added and incubated at 37 °C for 60 min. TMB (Merck, Darmstadt, Germany) was added as substrate, and the reaction was stopped with 1 M sulfuric acid and the plates read at an optical density (OD) of 450 nm (SpectraMax ABS Plus, Molecular Devices, San Jose, CA, USA). We used a 7-step serially diluted SC-positive serum pool as the standard curve to recalculate the OD values into concentrations. All the samples were analysed in duplicate and reanalysed if the coefficient of variation (CV) between the duplicates was >20%. The inter-assay CV was 9%, and the intra-assay CV was 2%, respectively.

The serum SC ACPAs were measured by modifying anti-CCP2 ELISA kits (CCPlus^®^ Immunoscan; Svar Life Science, Malmö, Sweden). In brief, the serum samples were diluted 1:25 in kit buffer, and the secondary antibody, detecting human secretory component, was diluted 1:2000 (polyclonal goat antibody conjugated to horseradish peroxidase, GAHu/SC/PO; Nordic Biosite, Täby, Sweden). Incubation and washing were performed according to the manufacturer’s instructions. The reaction was stopped and the plates read at OD_450nm_ (SpectraMax ABS Plus). A 7-step standard curve based on a serum sample with known high levels of SC ACPAs was used to recalculate the OD values into arbitrary units. All the samples were analysed in duplicate and reanalysed if the CV between the duplicates was >20%. The inter-assay CV was 10%, and the intra-assay CV was 5%.

The presence of antibodies (IgG) against the oral pathogen *P. gingivalis* virulence factor arginine gingipains (Rgp) was assessed using an in-house ELISA, using the RgpB protein, purified from *P. gingivalis* cultures, as the coating antigen, as previously described [9]. The samples were analysed in duplicate, and a standard curve (pool of Rgp IgG-positive sera) was used to present the antibody levels in arbitrary units (AU).

The total levels of IgA, IgA1 and IgA2 were measured using in-house ELISAs. The total IgA ELISA utilised F(ab′)2 fragment goat anti-alpha chain antibodies for capture and detection (Jackson ImmunoResearch, West Grove, PA, USA), and the subclass-specific ELISAs both used the same capture antibody as for total IgA and the detection antibodies ABIN135642 for IgA1 and ABIN135646 for IgA 2, respectively (www.antibodies-online.com, accessed on 19 February 2024). The same normal serum was used as the standard curve for all the analyses; the levels of total IgA, IgA1 and IgA2 were determined at the Uppsala University laboratory.

### 2.7. Statistical Analysis

All the statistical analyses were performed using SPSS v. 28 for Windows (SPSS Inc., Chicago, IL, USA). The data tested for normality using the Shapiro–Wilk test are expressed as means ± SD and the skew-distributed data as medians and lower and upper quartiles. An independent t-test and the Mann–Whitney U test were used to compare current smokers and non-smokers. For the frequency distribution, Pearson’s χ^2^-test was used. Correlations were tested using Spearman’s rank order correlation. A *p*-value of <0.05 was considered significant. The significance levels are presented both without and with Bonferroni correction for the number of NO variables investigated.

## 3. Results

### 3.1. Baseline Characteristics of the Study Population

As smoking affects pulmonary NO [28], the study participants (n = 44), 26 females and 18 males, were divided into subgroups depending on their current smoking status. The smoking subjects had significantly lower FE_NO,50_ and CA_NO_ values but higher free SC levels in their serum (Table 1). A total of 10 subjects could not perform the lowest exhalation flow, and the results for Caw_NO_ and Daw_NO_ are given for 34 subjects in Table 1.

### 3.2. NO in Relation to Clinical and Inflammation Markers

The serum levels of NO were correlated weakly with FE_NO,50_ (Rho 0.304, *p* = 0.048). As in our previous study, we found no association with DAS28, but when splitting DAS28 into separate components (swollen joints, tender joints, general health and ESR), we found a negative correlation between the number of swollen joints and FE_NO,50_ and especially Caw_NO_ (Table 2 and Figure 1A,B). For Daw_NO_, this correlation was positive (Table 2). CA_NO_ was correlated with both CRP and SC (Table 2).

### 3.3. NO in Relation to Antibodies

In this material, of the anti-CCP-IgG-positive RA participants, 52% also had anti-CCP2 IgA, 93% had IgM RF and 64% had IgA RF. The levels of anti-CCP2 (IgA and IgG) and IgM RF did not correlate with the exhaled NO parameters (Table 2). IgA RF was correlated negatively with Caw_NO_ and FE_NO,50_, while Daw_NO_ showed a positive correlation (Table 2 and Figure 1C,D). Among the smokers, the proportion of IgA-RF-positive parties was 83% compared with 67% in the non-smokers (n.s.). When comparing the non-smoking patients with the current smokers, there were no statistical differences in any antibody level.

Free SC in the serum was found in 44% of patients (67% current smokers, 36% non-smokers, *p* = 0.065), and its levels were negatively correlated with CA_NO_. The SC ACPAs, anti-Rgp IgG and the total levels of the IgA and IgA subclasses did not correlate with the exhaled NO parameters (Table 2).

## 4. Discussion

The main findings in the present study were negative correlations between the exhaled NO levels, especially the airway wall concentration of NO (Caw_NO_), and the number of swollen joints and IgA RF levels, respectively, in ACPA-positive RA. In addition, the lack of an association with the anti-CCP2 antibody levels or the SC ACPAs suggests that altered NO dynamics in the lungs reflect another biological process in early RA.

We have previously found that patients with recent-onset RA have significantly lower exhaled NO parameters than matched healthy control subjects [18]. It is known that smoking decreases these levels in healthy individuals. Our studies confirm that this is true for patients with early and chronic RA as well [18,19]. However, the levels were also low in the non-smoking patients, especially Caw_NO_, suggesting that smoking reduces the already low NO levels in the airways of ACPA-positive RA patients. It is not known whether other inhaled agents associated with RA pathogenesis, such as silica and textile dust, may also reduce the exhaled NO levels in non-smoking RA patients.

As smoking is a known triggering factor in the pathogenesis of seropositive RA [29], we looked for an association between RA-specific autoantibodies and the exhaled NO levels. We found a significant negative correlation between IgA RF and the NO levels in the airway wall compartment but not for the NO levels in the alveoli, suggesting an immune-mediated process localised to the airways. The link between airway diseases and ACPA-positive RA is excellently described by Matson et al. [30]. The authors also present theories on the immune reactions of the airways and their relevance to RA’s pathogenesis.

Generally, IgA is a secretory antibody dominating the mucosal tissue. Smoking in healthy individuals may increase serum IgA levels [31], including IgA RF, as shown in a non-arthritic population from Iceland, where the presence of IgA RF was more frequent in smokers than in non-smokers [32]. Also, among RA patients, smokers also have elevated IgA RF levels compared to non-smokers [31], and we saw the same trend in our early RA cohort. Additionally, smoking has been linked to increased RA disease activity [33] and a reduced RA treatment response to anti-rheumatic therapies, such as anti-TNF treatment [34].

In patients with RA, serum IgA RF levels correlate with saliva and tear fluid levels, which may support local mucosal IgA RF production [35]. In our study, IgA RF was correlated negatively with Caw_NO_ and FE_NO,50_, and since IgA antibodies are present mainly in the mucosa, our data support the hypothesis that immune reactions in the airway mucosa are essential in early seropositive RA.

RA patients have elevated levels of total IgA [36,37], and serum levels of IgA and IgG can predict future RA development [38], while serum levels of IgA, but not IgG or IgM, have been associated with the degree of cartilage erosions in patients with established RA for ≥1 year [39]. Due to the unique association between IgA RF and the NO parameters in our study, pointing towards primarily airway involvement, we also analysed the total levels of IgA as well as the IgA subclasses in relation to the NO parameters (Table 2) and smoking (Table 1), but we did not find any associations.

In line with previous findings [8], the free SC in the serum was increased among the currently or previously smoking RA patients. Therefore, the negative correlation between CA_NO_ and free SC in the serum is difficult to disentangle since smoking is associated with reduced NO levels. Nevertheless, the poly Ig receptor is readily expressed in small airways [40], and RA-related pro-inflammatory cytokines such as interferon gamma and TNF are known to upregulate poly Ig receptor expression [41]. Thus, a potential link between local NO production, inflammation and free SC release may exist but needs further characterisation [41].

We found no correlation between the exhaled NO parameters and the ACPA levels, measured as IgG and IgA anti-CCP2 or SC ACPAs. We deliberately focused this investigation on ACPA-positive RA, as this is the subgroup of patients with the most severe prognosis and most probably represents a disease entity on its own [42]. Therefore, we do not know how these analyses would have looked if we had included both (IgG) ACPA-positive and ACPA-negative patients, nor did we find any associations between the NO levels and antibodies against the *P. gingivalis* antigen Rgp. The low NO levels in the exhaled gas may reflect NO-consuming processes in RA pathogenesis, such as oxidative stress during inflammation and the production of reactive oxygen species [43]. Alternatively, this may depend on arginine depletion because of the increased arginase activity seen in RA and the subsequent substrate competition with inducible nitric oxide synthase (iNOS) [44]. Both enzymes use arginine as a substrate for their activity.

One or more swollen joints are required for RA diagnosis [20], and this clinical finding is essential to and the basis for all arthritis diagnoses. The strong correlation observed between the number of swollen joints and decreasing exhaled NO parameters in early ACPA-positive RA, which remained after Bonferroni correction for multiple testing, supports the need for future studies on exhaled NO as a predictor of arthritis onset in ACPA-positive at-risk individuals. This is further corroborated by our own recent findings, showing that in early RA, ACPAs but not RF are specifically associated with a lower number of inflamed joints as compared to ACPA-negative patients [45]. Together, these findings implicate that in early ACPA-positive RA, a low number of swollen joints is strongly associated with a higher airway NO concentration (but still lower than that of healthy controls) [18] and that this local change in NO in the airway walls might be specifically associated with ACPA-positive RA. Further studies regarding exhaled NO in ACPA-positive individuals who subsequently develop RA are therefore needed. What impacts the size of the swollen joints and the NO levels in synovial fluid may have has not been investigated either. The exhaled NO levels in other arthritis diseases have also been incompletely studied. Several questions remain to be answered on this topic.

Our findings of reduced exhaled NO parameters in association with IgA RF levels—but not IgM RF or ACPA levels—may indicate that several different inflammatory and immunological reactions are present in the airway mucosa of patients with early RA. The lack of an association with ACPA levels may suggest that our findings reflect another parallel and independent smoking-related immunological process. Moreover, the lack of an association between NO and anti-Rgp IgG levels (reflecting periodontitis) suggests that periodontal inflammation, which is linked to ACPA-positive RA, does not influence the exhaled NO levels.

A limitation of this study is the small number of participating subjects, and a larger study is needed to confirm the results. In addition, parallel examinations of ACPA-positive and ACPA-negative RA patients, and high-resolution computed tomography of the lungs to detect parenchymal changes, would be desired. As interstitial lung disease is common but present only in a minority of patients with early RA [46], a future investigation correcting for these limitations should be a larger multicentre study.

In conclusion, the altered NO dynamics of the lungs in patients with early ACPA-positive RA were correlated with IgA RF levels and the number of swollen joints, suggesting that NO, especially a reduced NO concentration in the airway walls, may be relevant to RA pathogenesis.

## Figures and Tables

**Figure 1 biomedicines-12-00964-f001:**
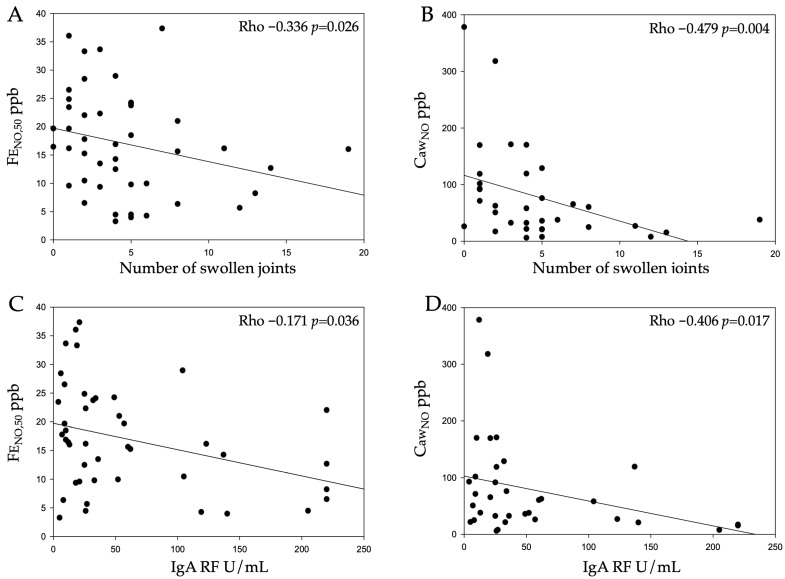
Correlation plots for FE_NO,50_ and Caw_NO_ for the number of swollen joints (**A**,**B**), and for IgA RF (**C**,**D**).

**Table 1 biomedicines-12-00964-t001:** Characteristics of participants with recent-onset ACPA-positive RA.

	All (n = 44)	Non-Smokers (n = 32)	Current Smokers (n = 12)	*p*-Value
Age (years)	60 ± 14	59 ± 16	62 ± 9	0.521
Sex (% female)	59%	66%	42%	0.150
BMI	28 ± 5	28 ± 5	28 ± 4	0.665
Symptom duration (months)	4 (2, 8)	5 (3, 8)	3 (2,11)	0.195
DAS28	4.47 ± 1.06	4.40 ± 1.19	4.65 ± 0.59	0.495
- Swollen joints	4 (2, 6)	3 (2, 5)	5 (3, 8)	0.153
- Tender joints	4 (2, 5)	4 (2, 6)	4 (1, 5)	0.866
- Global health	45 ± 24	42 ± 25	50 ± 21	0.478
- ESR	24 (12, 42)	20 (12, 40)	30 (13, 42)	0.679
CRP	7.9 (3.2, 20)	8.1 (2.6, 23)	7.1 (3.3, 14)	0.576
NOx	2.3 (1.8, 3.1)	2.4 (1.6, 3.1)	2.1 (1.3, 3.0)	0.243
HAQ	0.92 ± 0.46	0.89 ± 0.47	0.98 ± 0.45	0.591
*Immunological markers*				
IgA anti-CCP2 AU/mL	10.5 (3.1, 23)	6.8 (3.6, 21)	15.5 (2.4, 28)	0.668
IgG anti-CCP2 AU/mL	299 (93, 600)	288 (122, 600)	527 (56, 600)	0.706
IgA RF AU/mL	26 (12, 94)	25 (10, 56)	56 (25, 136)	0.059
IgM RF IU/mL	62 (20, 179)	49 (18, 131)	165 (34, 250)	0.100
Anti-Rgp AU/mL	361 (174, 727)	348 (168, 766)	369 (175, 563)	0.969
SC	0 (0, 30)	0 (0, 13)	24 (0, 47)	**0.035**
SC ACPAs	29 (12, 130)	24 (11, 62)	119 (16, 217)	0.082
Total IgA g/L	4.72 (3.77, 6.07)	4.76 (3.78, 6.31)	4.72 (3.85, 6.07)	0.645
Total IgA1, g/L	4.15 (2.74, 7.18)	4.15 (2.75, 7.18)	4.32 (2.56, 9.25)	1.000
Total IgA2, g/L	0.81 (0.48, 1.25)	0.78 (0.45, 1.23)	0.90 (0.54, 1.66)	0.377
*NO analysis*				
FE_NO,50_ ppb	16 (10, 24)	19 (13, 25)	10 (5, 16)	**0.002**
CA_NO_ ppb	1.6 (1.0, 2.2)	1.9 (1.2, 2.3)	1.0 (0.4, 1.3)	**0.004**
Caw_NO_ ppb (n = 34)	55 (24, 106)	64 (33, 115)	24 (20, 75)	0.086
Daw_NO_ mL/s (n = 34)	17 (8, 30)	16 (9, 31)	19 (7, 29)	0.809
*Lung function*				
FEV_1_ % predicted	84 ± 15	87 ± 15	78 ± 15	0.446
FVC % predicted	86 ± 11	87 ± 11	84 ± 13	0.071

Data presented as means ± SD, and for skewed data, as medians (25–75-percentile); *p*-value of <0.05 was considered significant (highlighted in bold). BMI, body mass index; DAS28, Disease Activity Score for 28 joints; ESR, erythrocyte sedimentation rate; CRP, C-reactive protein; NOx, serum nitrate/nitrite; HAQ, the Swedish version of the Stanford Health Assessment Questionnaire; RF, rheumatoid factor; ACPA, anti-citrullinated protein antibody; anti-CCP2, antibody against cyclic citrullinated peptide, second generation; anti-Rgp; antibody against *P. gingivalis* arginine gingipains; IgA/IgG/IgM, immunoglobulin isotypes A, G and M; SC, secretory component; SC ACPAs, SC ACPA immune complex; FE_NO,50_, fraction of exhaled nitric oxide at a flow of 50 mL/s; CA_NO_, alveolar nitric oxide; Caw_NO_, nitric oxide content in the airway wall; Daw_NO_, nitric oxide diffusion capacity over airway wall; FEV_1_, forced expiratory volume at 1 s; FVC, forced vital capacity.

**Table 2 biomedicines-12-00964-t002:** Correlation of FE_NO,50_ and NO parameters representing the airways and lungs in all participants.

	FE_NO,50_ (n = 44)		Caw_NO_ (n = 34)		Daw_NO_ (n = 34)		CA_NO_ (n = 44)	
	Rho	*p*-Value	Rho	*p*-Value	Rho	*p*-Value	Rho	*p*-Value
*DAS28*								
Total score	−0.248	0.109	−0.327	0.063	0.150	0.406	0.169	0.279
Number of swollen joints	−0.336	**0.026**	−0.479	**0.004 ***	0.385	**0.025**	0.023	0.882
Number of tender joints	−0.171	0.268	−0.140	0.430	−0.076	0.669	−0.025	0.870
Global Health	−0.130	0.406	−0.023	0.901	−0.129	0.473	−0.068	0.666
ESR	−0.001	0.995	−0.111	0.532	0.167	0.346	0.285	0.064
CRP	0.030	0.848	−0.136	0.442	0.172	0.332	0.362	**0.016**
*Immunological markers*								
IgA anti-CCP2	−0.113	0.465	−0.180	0.309	0.138	0.435	0.063	0.686
IgG anti-CCP2	−0.148	0.311	−0.181	0.305	0.107	0.547	−0.027	0.860
IgA RF	−0.171	**0.036**	−0.406	**0.017**	0.357	**0.037**	−0.001	0.994
IgM RF	−0.173	0.262	−0.151	0.394	0.001	0.994	0.039	0.804
IgG anti-Rgp	−0.063	0.683	−0.044	0.803	0.027	0.878	0.015	0.922
SC	−0.173	0.266	−0.007	0.971	−0.160	0.366	−0.345	**0.023**
SC ACPA	−0.255	0.103	−0.050	0.783	−0.093	0.608	−0.131	0.408
Total IgA	0.184	0.231	0.087	0.624	0.141	0.427	0.071	0.646
Total IgA1	0.270	0.076	−0.035	0.844	0.242	0.168	0.128	0.406
Total IgA2	0.062	0.689	−0.169	0.339	0.130	0.462	−0.047	0.759

* = remains significant (*p* = 0.020) after Bonferroni correction. *p*-value of <0.05 was considered significant (highlighted in bold). DAS28, Disease Activity Score for 28 joints; ESR, erythrocyte sedimentation rate; CRP, C-reactive protein; RF, rheumatoid factor; ACPA, anti-citrullinated protein antibody; anti-CCP2, antibody against cyclic citrullinated peptide, second generation; anti-Rgp, antibody against *P. gingivalis* arginine gingipains; IgA/IgG/IgM, immunoglobulin isotypes A, G and M; SC, secretory component; SC ACPAs, SC ACPA immune complex; FE_NO,50_, fraction of exhaled nitric oxide at a flow of 50 mL/s; CA_NO_, alveolar nitric oxide; Caw_NO_, nitric oxide content in the airway wall; Daw_NO_, nitric oxide diffusion capacity over airway wall.

## Data Availability

Data are available upon reasonable request by contacting the corresponding author.

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
