# Peer review of "Exhaled Nitric Oxide Reflects the Immune Reactions of the Airways in Early Rheumatoid Arthritis"

_biomedicines, 2024, doi:10.3390/biomedicines12050964_

Round 1
Reviewer 1 Report
Comments and Suggestions for Authors
The aim of the reviewed paper is to provide some insights into the role of evaluation of exhaled nitric oxide in early rheumatoid arthritis (RA). Even if the subject is not one extensively explored in RA, till now there are already several data related to this subject. Also, by reading this article and another one by the same authors published in 2022 doi: 10.1038/s41598-022-10334-5, I observed several similarities (including entire sentences) that could explain a quite high percent match of 32% on iThenticate report – please revise this aspect!
Some other comments:
-row 46 – since RA does not affect only the joints I would not use the term of “inflammatory joint disease”
-rows 50-51 – please provide references for this statement
-row 83 – the sentence is quite unclear, please reformulate it
- rows 88-89 – you mention study population as “early-onset RA” – how was it defined, what was considered “early-onset” for your cohort?
What about Local Ethical Committee approval?
- What type of DAS28 was used (CRP or ESR)?
Results are mainly presented by using two tables (no 1 & 2) but they should be also extensively explained – it is quite difficult to follow like this.
Finally, what is the actual utility of the presented study? Did you try to compare this early cohort (apparently without interstitial involvement) with a more late one, possibly with interstitial involvement in order to identify a possible utility of evaluation of exhaled NO in RA? Please expand final conclusions!
Author Response
Reviewer 1
We would like to thank the reviewer for an excellent and careful review.
The aim of the reviewed paper is to provide some insights into the role of evaluation of exhaled nitric oxide in early rheumatoid arthritis (RA). Even if the subject is not one extensively explored in RA, till now there are already several data related to this subject. Also, by reading this article and another one by the same authors published in 2022 doi: 10.1038/s41598-022-10334-5, I observed several similarities (including entire sentences) that could explain a quite high percent match of 32% on iThenticate report – please revise this aspect!
The two studies are from the same patient cohort, and this study is an extension of the first one. This means that patient background, recruitment, and study performance should be described similarly in the two papers. We have tried to vary and change the text as much as possible.
-row 46 – since RA does not affect only the joints I would not use the term of “inflammatory joint disease”
We agree to change the expression ”inflammatory joint disease” to a ”systemic inflammatory disease , mainly affecting the joints”.
-rows 50-51 – please provide references for this statement
A reference from Klareskog et al. is added
-row 83 – the sentence is quite unclear, please reformulate it
The sentence is changed and now reads: These subjects representing a homogenous autoimmune disease entity all with ACPA antibodies, were re-analysed in the present study, where the objectives were to elucidate if exhaled NO levels were associated with characteristicRA autoantibodies and other markers for the autoimmune process.
- rows 88-89 – you mention study population as “early-onset RA” – how was it defined, what was considered “early-onset” for your cohort?
Early RA is defined as symptom duration less than two years and fulfilling 2010 criteria for RA (described in the Method section under ”Study population”)
What about Local Ethical Committee approval?
Ethical approval is described on Row 334. Sweden does not have a local review board.
What type of DAS28 was used (CRP or ESR)?
In the DAS28 analysis, ESR was used in accordance with reference 20. The DAS28 is split into separate components, which are also shown in Table 1. The CRP levels are presented separately in Table 1.
Results are mainly presented by using two tables (no 1 & 2) but they should be also extensively explained – it is quite difficult to follow like this.
We have tried to have clear tables and refer to them in the text where we explain the results.
Finally, what is the actual utility of the presented study? Did you try to compare this early cohort (apparently without interstitial involvement) with a more late one, possibly with interstitial involvement in order to identify a possible utility of evaluation of exhaled NO in RA? Please expand final conclusions!
Our interest in exhaled NO started with our findings of low levels of exhaled NO in a cross-sectional study of chronic RA patients (Ref 18 ), but in that study and this one, we did not have HRCT (high-resolution computer tomography) of the lungs, so we do not know the presence of interstitial lung disease in this early RA cohort. This is described as a limitation of the study (Row 307). We have expanded the conclusion to describe that future studies should include early RA patients with ILD, but as this is not present in all patients, that will require a much larger study on many centra and, ideally, also investigations on pre-RA patients. But as with all research, that will lead to another study.
Reviewer 2 Report
Comments and Suggestions for Authors
A manuscript (ID: biomedicines-2923963) entitled “Exhaled nitric oxide reflects immune reactions of airways in early rheumatoid arthritis” has been submitted to the scientific journal biomedicines. The draft documents a solid piece of scientific work and should fit in a perfect way to the topics discussed in this journal. Therefore, I have only minor remarks. Tomas Weisoft et al. have demonstrated that a relationship between exhaled NO and rheumatoid arthritis exist. I suggest, if possible, that the authors further try to depict the relationship between NO and rheumatoid arthritis through animal experiments to better understand the pathogenesis of rheumatoid arthritis. Furthermore, the manuscrift could benefit from a discussion of this publication:
S. M. Matson, M. K. Demoruelle, M. Castro (2022) Airway Disease in Rheumatoid Arthritis Ann. Am. Thorac Soc.19(3), 343–352. DOI: 10.1513/AnnalsATS.202107-876CME
Author Response
A manuscript (ID: biomedicines-2923963) entitled “Exhaled nitric oxide reflects immune reactions of airways in early rheumatoid arthritis” has been submitted to the scientific journal biomedicines. The draft documents a solid piece of scientific work and should fit in a perfect way to the topics discussed in this journal. Therefore, I have only minor remarks. Tomas Weisoft et al. have demonstrated that a relationship between exhaled NO and rheumatoid arthritis exist. I suggest, if possible, that the authors further try to depict the relationship between NO and rheumatoid arthritis through animal experiments to better understand the pathogenesis of rheumatoid arthritis. Furthermore, the manuscrift could benefit from a discussion of this publication:
- M. Matson, M. K. Demoruelle, M. Castro (2022) Airway Disease in Rheumatoid Arthritis Ann. Am. Thorac Soc.19(3), 343–352. DOI: 10.1513/AnnalsATS.202107-876CME
As far as we know, there is no applicable animal model for ACPA-positive RA .
Studies of exhaled NO in animals can only be done using a method of collecting all exhaled gases. The NO dynamics of the lung require a lot of cooperation with the investigated person (see Method section).
The suggested reference is excellent and included in the reference list and discussed briefly.
Round 2
Reviewer 1 Report
Comments and Suggestions for Authors
Thank you for your update. Actual form of the article responded to all my suggestions.